# Three-year hospital-wide pain management system implementation at a tertiary medical center: Pain prevalence analysis

**Ming-Chuan Chen**[1,2], **Te-Feng Yeh**[3], **Chih-Cheng Wu**[1,4,5], **Yan-Ru Wang**[3], **Chieh-Liang Wu**[6], **Ruei-ling Chen**[7], **Ching-Hui Shen**[1,8,9]*

**1** Department of Anesthesiology, Taichung Veterans General Hospital, Taichung, Taiwan, **2** Department of Anesthesiology, Chiayi Branch, Taichung Veterans General Hospital, Chiayi, Taiwan, **3** Department of Healthcare Administration, Central Taiwan University of Science and Technology, Taichung, Taiwan, **4** Department of Financial Engineering, Providence University, Taichung, Taiwan, **5** Department of Data Science and Big Data Analytics, Providence University, Taichung, Taiwan, **6** Center of Smart Healthcare, Taichung Veterans General Hospital, Taichung, Taiwan, **7** Department of Nursing, Taichung Veterans General Hospital, Taichung, Taiwan, **8** School of Medicine, National Yang Ming Chiao Tung University, Taipei, Taiwan, **9** Department of Post-Baccalaureate Medicine, College of Medicine, National Chung Hsing University, Taichung, Taiwan

* shench07@gmail.com

**Data Availability Statement:** All relevant data are within the paper and its Supporting information files.

## Abstract

We developed a pain management system over a 3-year period. In this project, "Towards a pain-free hospital", we combined evidence-based medicine and medical expertise to develop a series of policies. The intervention mainly included the development of standard procedures for inpatient pain management, the implementation of hospital-wide pain medicine education and training, the establishment of a dashboard system to track pain status, and regular audits and feedback. This study aimed to gain an understanding of the changes in the prevalence of pain in inpatients under the care of the pain management system. The subjects of the survey are inpatients over 20 years old, and who had been hospitalized in the general ward for at least 3 days. The patients would be excluded if they were unable to respond to the questions. We randomly selected eligible patients in the general ward. Our trained interviewers visited inpatients to complete the questionnaires designed by our pain care specialists. A total of 3,094 inpatients completed the survey from 2018 to 2020. During the three-year period, the prevalence of pain was 69.5% (2018) (reference), 63.3% (2019) (OR:0.768, p<0.01), and 60.1% (2020) (OR:0.662, p <0.001). The prevalence rates of pain in patients undergoing surgery during the 3-year period were 81.4% (2018), 74.3% (2019), and 68.8% (2020), respectively. As for care-related causes of pain, injection, change in position/chest percussion, and rehabilitation showed a decreasing trend over the 3-year period of study. Our pain management system provided immediate professional pain management, and achieved a good result in the management of acute moderate to severe pain, especially perioperative pain. Studies on pain prevalence and Pain-Free Hospitals are scarce in Asia. With the aid of the policies based on evidence-based medicine and the dashboard information system, from 2018 to 2020, the prevalence of pain has decreased year by year.

**Funding:** The author(s) received no specific funding for this work.

**Competing interests:** The authors have declared that no competing interests exist.

## Introduction

Pain is one of the most common complaints of inpatients in the hospital. The prevalence of pain in hospitals in various countries ranges from 52 to 100% [1–4]. We conducted a study of pain prevalence among inpatients in 2018 [5], and nearly 70% of hospitalized patients experienced pain. Therefore, pain management is an important issue in all medical institutions. Pain-related stress may cause instability in vital signs such as blood pressure and heart rate. It can also worsen a patient's quality of life, and has a deleterious effect on recovery from illness or surgery [6]. Poor pain management leads to longer hospital stays and increased readmissions, resulting in a waste of medical resources [7]. The hospital should provide comprehensive pain management to improve the physical and mental condition of most patients suffering from pain.

"Towards a Pain-Free Hospital" was first started in Montreal, Canada [8]. The features of the campaign are as following. (1) Pain is a problem common to all medical specialities. (2) Every hospital professional is faced with the problem of pain. (3) The whole hospital must be involved in the campaign. (4) The general public should be involved in the campaign as well. It has been implemented in some hospitals in North America and Europe, and its efficacy has also been evaluated. Twenty hospitals in Italy participated in this campaign in 2000 [9]. In Germany the campaign was launched in 2003 to improve the effectiveness of pain management. It was found that 55% of German surgical patients and 57% of non-surgical patients were not satisfied with their pain management. The patients believed that effective pain management remained insufficient [10]. The evidence shows pain management does not meet the expectations of hospitalized patients. In addition to developing and updating clinical care guidelines for pain management [11, 12], it is necessary to develop an effective strategy to overcome the wide gap between clinical care guidelines and real-life care processes.

Evidence-based practices can be achieved with continuous quality management. Auditing and feedback generally lead to small but potentially important improvements in professional practice [13]. Introducing clinical and quality dashboards can have a positive effect on care outcomes and processes of care [14]. Needham DM proposed a model for undertaking knowledge translation: step 1, summarize the evidence; step 2, identify local barriers to implementation; step 3, measure performance; and step 4, ensure that patients have received the intervention [15]. In terms of hospitalized pain management, it is necessary to translate the knowledge of pain management step by step into real-life care with the Plan-Do-Check-Act (PDCA) procedure and conducting effective auditing and feedback using the indicators shown on the dashboard.

To reach the goal of "a pain-free hospital", we launched a hospital-wide pain management project in 2018. The project adopted an evidence-based approach with a data-driven information system to control pain. We surveyed patients' pain prevalence from 2018 to 2020. A detailed description of the project and the methods that were used to successfully improve pain management among the hospitalized patients is provided below.

## Methods

### Study setting

Taichung Veterans General Hospital is a 1500-bed hospital and medical center located in central Taiwan. There are 19 general wards in our hospital, with about 50,000 hospitalizations per year. There is no dedicated ward for pain patients. We implemented a hospital-wide pain management campaign in 2018, and surveyed the pain prevalence of inpatients from 2018 to 2020.

This research was approved by the Institutional Review Board I & II of Taichung Veterans General Hospital (Certificate Number: CE18236B).

## The project of hospital-wide pain management

The purpose of this project is to comprehensively relieve pain and improve medical quality. We are committed to (1) using the same tool or technique for all medical staff to assess pain, at least 3 times a day, and actively treat pain, (2) documenting pain management in detail, (3) implementing pain medical education for all medical staff in the hospital and (4) monitoring the comorbidities associated with pain management. The initial preparation for the pain management project in our hospital commenced in 2017. First, we organized pain management committees and recruited specialists in pain management, which included a nursing care team, an education team, computer technicians, a quality improvement team, and the leadership (S1 Fig). The "Towards a Pain-Free Hospital" team has 28 members in total, including the convener, deputy convener, 19 physicians, 3 nurses, 3 administrative staff and 1 information engineering staff. Second, we searched for literature on Towards a Pain-Free Hospital and hospital-wide pain management according to the hospital environment and conditions. We reviewed the recent literature to update our guidelines for pain management [16–22]. Third, we developed teaching materials based on our guidelines and launched the hospital-wide education program in May, 2018. We trained the seed instructors to provide lectures for each division and each ward. The training courses and lectures in each unit were repeated every year. We also provided feedback on the achievement of each unit during the lecture. Fourth, we designed our daily pain indicator "3324" which is detailed in the following section. Our computer and medical informatics engineers designed the pain management system and dashboard (S2 Fig), which integrate the nurses' measurements of the patients' pain levels, pain medications, and the alarm of 3324. Fifth, using the plan-do-check-act (PDCA) concept, the indicators of the process and outcome in the pain management were reviewed by the team of the participating wards, divisions, committees, and the hospital at the indicated periods (S1 Table).

## The response cycle of the real-time pain indicator of "3324"

According to National Comprehensive Cancer Network (NCCN) guideline, to allow for dose titration, the short-acting opioids could be given as often as once per hour as needed, if pain is inadequately controlled. If hourly dosing is needed for more than 3 cycles, reassessment or other intervention is recommended. Therefore, we believe that the occurrence of breakthrough pain in patients should be controlled at less than 3 times a day. In addition, the patient's pain should be managed at a near-painless level (Visual Analogue Scale, VAS $\leq$ 3) with minimal side effects. Based on this, we had formulated the "3-3-3 principle" for chronic pain, which means that the patient's pain should be controlled to a VAS of 3 or less within 3 days of admission, and the frequency of breakthrough pain should be less than 3 times a day. In recent years, in response to the Towards a Pain-Free Hospital policy, considering the resources of the hospital and the feasibility of the policy, we hoped to develop a set of methods to cooperate with the dashboard system to monitor and identify patients with difficulty in pain management. Thus, we referred to the 3-3-3 principle and designed an approach termed "3324" for all pain monitoring in the hospital. This number means that the scale of VAS was $> 3$ and occurred $\geq 3$ times within the last 24 hours during hospitalization. The data of patients who meet the criteria of 3324 will be displayed in the Dashboard Pain Management System, and the nurses and physicians can see it on the working panel of the electronic medical record system. The medical staff then take the necessary actions to control the patients' pain following the process (Fig 1).

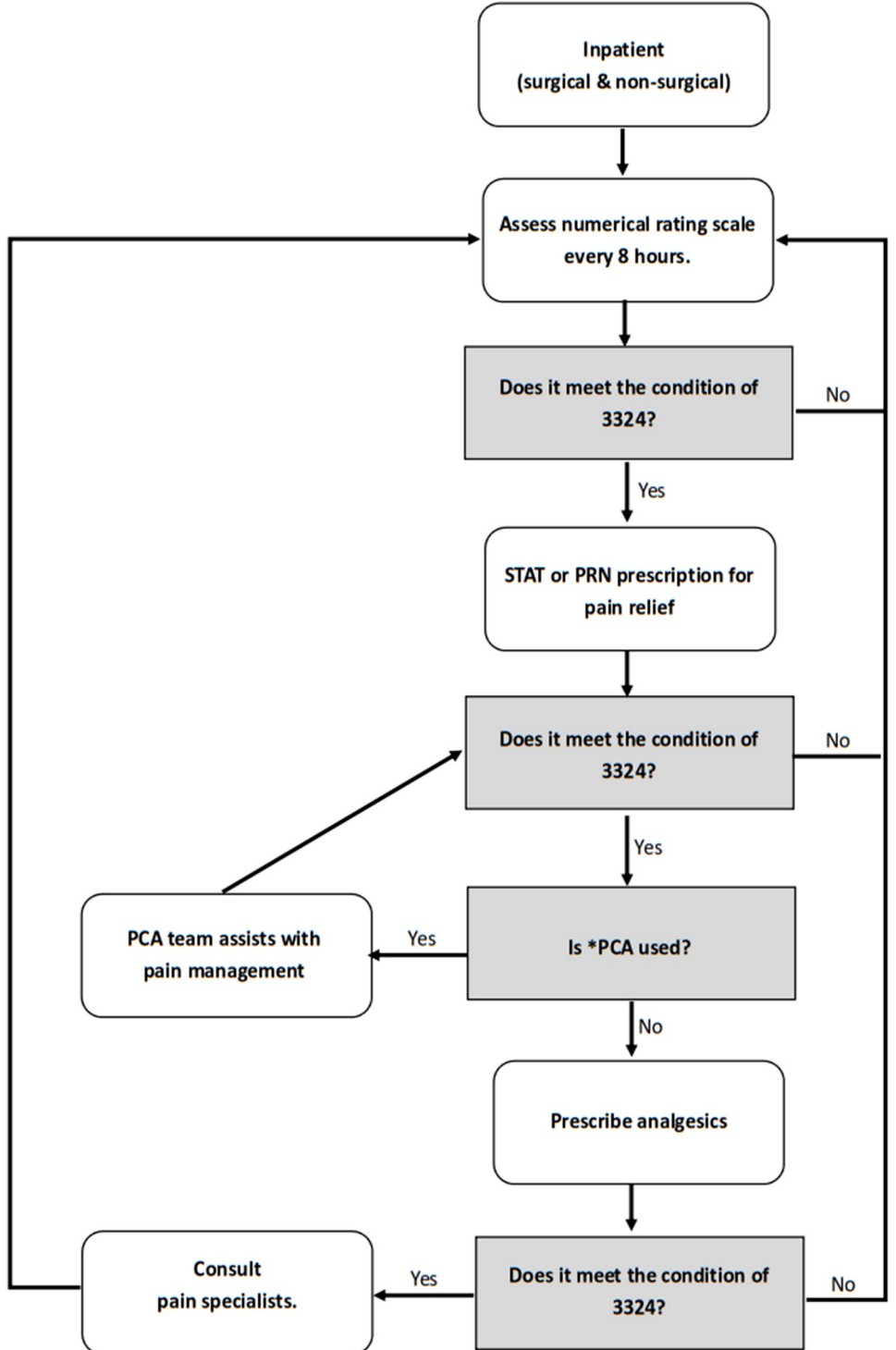

**Fig 1. Standard process for inpatient pain management.** Based on clinical care guidelines and evidence-based medicine, we developed the "Standard Process for Inpatient Pain Management", and set up quality indicators from pain management process for auditing and feedback [16–19]. *PCA: patient-controlled analgesia.

If the patient still has refractory pain, the case manager of pain (a nurse) can visit the patient and consult with a pain specialist (anesthesiology) to adjust the analgesic prescription or consider regional analgesic therapies, such as peripheral nerve block. We hope to use this to identify inpatients who have difficulty in pain management immediately, and consult pain physicians to provide individualized treatment based on the patient's condition, so as to improve the quality of pain management in the hospital. For example, a patient suffers from postoperative wound pain. If the ward nurse assesses that the pain level exceeds 3 points, analgesics will be given according to the postoperative doctor's order of the surgeon, and guide the patient how to use the PCA pump. If the analgesic effect is poor, the PCA team of the pain department will be asked to adjust the dosage of the pump, or the ward physician will be asked to evaluate the situation and prescribe an analgesic drug. If the above treatments are still ineffective, or if the pain level is greater than three times within 24 hours, a pain specialist will be consulted to the ward to evaluate the pain and provide advice. Cases that meet 3324 will be automatically tracked in the dashboard pain management system through electronic records.

## Pain prevalence survey

We surveyed the pain prevalence of inpatients from 2018 to 2020, according to the method described in our previous research [5]. We used a questionnaire that is divided into 4 main parts, including (1) personal information, (2) pain experience during hospitalization, (3) perception of pain care from physicians and nurses, and (4) overall satisfaction (S1 Appendix). All inpatients in general wards of Taichung Veterans General Hospital were the research subjects. The patients (1) were at least twenty years old, (2) had a hospital stay of at least three days, (3) had clear consciousness, and (4) provided consent to be included in this study. Patients were excluded if they were not able to express themselves e.g., had dementia, brain injury, or unstable medical condition, such as shock or coma. Five experts evaluated this questionnaire with a Content Validity Index (CVI) of 0.90. Before the questionnaire survey, 30 inpatients were sampled in the general ward for pretest. Regarding the reliability, we used KR20 and Cronbach's α coefficient to test the questionnaire. The interviewers conducted a face-to-face questionnaire interview in the ward. We recruited inpatients using quota sampling for each general ward. Enrollment will be stopped when each general ward reaches the quota sampling number of the day. To implement standardized interviews, we train interviewers before conducting the questionnaires. On ward visits, the nurses listed eligible patients, and we randomly sampled and interviewed patients. After the patients signed the informed consent document, we started our questionnaire.

## Statistics

All of the patient information was anonymized before analysis. We used numeric rating scale to classify pain levels into mild (1–3), moderate (4–6), and severe (7–10). We used the chi-square test to analyze the variability of the patient's demographic data and care-related pain with severity ≥ 4 and duration ≥ 4 hours. In addition, we used logistic regression to determine the factors related to pain and analyze trends in pain prevalence over the three years. An alpha level of $p < 0.05$ was considered statistically significant. All analyses were performed using SPSS Windows version 21.0.

## Results

### 3-year demographic data

To evaluate the outcome of the hospital-wide pain management project, we surveyed the patients' pain prevalence from 2018 to 2020 according to a previously published method [5].

**Table 1. Demographic data of the patients from 2018 to 2020.**

| Characteristics | ALL | 2018 | 2019 | 2020 | p-value |
|---|---|---|---|---|---|
| | N = 3,094 (%) | N = 1,034 (%) | N = 1,035 (%) | N = 1,025 (%) | |
| Gender | | | | | |
| Female | 1,415 (45.7) | 497 (48.1) | 458 (44.3) | 460 (44.9) | |
| Male | 1,679 (54.3) | 537 (51.9) | 577 (55.7) | 565 (55.1) | |
| Age (years) | | | | | <0.05 |
| ≤39 | 447 (14.1) | 168 (16.2) | 138 (13.3) | 141 (13.8) | |
| 40~49 | 374 (12.1) | 121 (11.7) | 119 (11.5) | 134 (13.1) | |
| 50~59 | 643 (20.8) | 240 (23.2) | 208 (20.1) | 195 (19.0) | |
| 60~69 | 789 (25.5) | 257 (24.9) | 258 (24.9) | 274 (26.7) | |
| 70~79 | 508 (16.4) | 136 (13.2) | 194 (18.7) | 178 (17.4) | |
| ≥80 | 333 (10.8) | 112 (10.8) | 118 (11.4) | 103 (10.0) | |
| Weight (kg) | | | | | <0.01 |
| ≤50 | 435 (14.1) | 144 (13.9) | 130 (12.6) | 161 (15.7) | |
| 50~59 | 874 (28.2) | 277 (26.8) | 291 (28.1) | 306 (29.9) | |
| 60~69 | 966 (31.2) | 322 (31.1) | 323 (31.2) | 321 (31.3) | |
| 70~79 | 470 (15.2) | 186 (18.0) | 166 (16.0) | 118 (11.5) | |
| ≥80 | 349 (11.3) | 105 (10.2) | 125 (12.1) | 119 (11.6) | |
| Education | | | | | |
| Illiterate | 215 (6.9) | 72 (7.0) | 73 (7.1) | 70 (6.8) | |
| Elementary school | 673 (21.8) | 220 (21.3) | 254 (24.5) | 199 (19.4) | |
| Junior high school | 460 (14.9) | 159 (15.4) | 155 (15.0) | 146 (14.2) | |
| Senior high school | 848 (27.4) | 284 (27.5) | 275 (26.6) | 289 (28.2) | |
| Bachelor's degree or higher | 898 (29.0) | 299 (28.9) | 278 (26.9) | 321 (31.3) | |
| Marriage | | | | | <0.01 |
| Unmarried | 397 (12.8) | 142 (13.7) | 122 (11.8) | 133 (13.0) | |
| Married | 2,380 (76.9) | 765 (74.0) | 803 (77.6) | 812 (79.2) | |
| Other | 317 (10.2) | 127 (12.3) | 110 (10.6) | 80 (7.8) | |
| Disease categories | | | | | |
| Surgery | 1,284 (41.5) | 400 (38.7) | 436 (42.1) | 448 (43.7) | |
| Head, Neck, Ophthalmology | 155 (5.0) | 55 (5.3) | 51 (4.9) | 49 (4.8) | |
| Internal Medicine | 1,464 (47.3) | 509 (49.2) | 485 (46.9) | 470 (45.9) | |
| Gynecology | 191 (6.2) | 70 (6.8) | 63 (6.1) | 58 (5.7) | |
| Operation | | | | | |
| No operation | 1,514 (48.9) | 490 (47.4) | 499 (48.2) | 525 (51.2) | |
| With operation | 1,580 (51.1) | 544 (52.6) | 536 (51.8) | 500 (48.8) | |

N (%): Data are expressed as the case number and its percentage among total cases.

The demographic data of all of the patients are shown in Table 1. In total, 1034, 1035, and 1025 patients were recruited by quota sampling from wards in 2018, 2019, and 2020, respectively. In 2020, we followed the same collection pattern as the previous year. However, because of COVID-19 outbreak, part of the ward was converted into a negative pressure isolation room. The total number of general beds was reduced by 10, so 1,025 samples were collected. Therefore, a total of 3,094 samples were collected in the three years, including 1679 males (54.3%) and 1415 females (45.7%). They were divided into six groups according to age, among which 789 patients (25.5%) were aged 60–69. They were divided into five groups according to body weight, among which 966 patients (31.2%) accounted for 60–69 kg. 898 patients (29.0%) had a

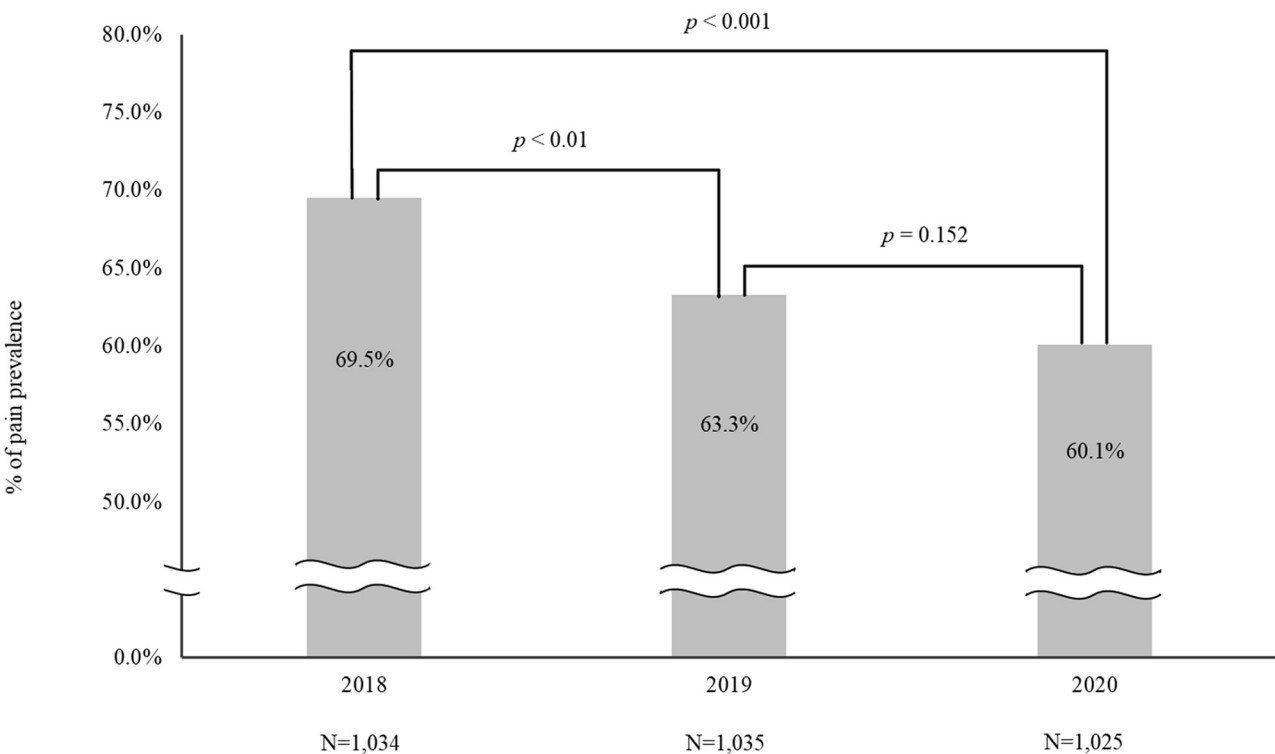

**Fig 2. Decrease of pain prevalence after implementing the hospitalized-wide pain management program from 2018 to 2020.**

college education or above, followed by senior high school (27.4%), elementary school (21.8%), junior high school (14.9%), and illiterate (6.9%). In terms of marital status, most of the 2380 (76.9%) were married, and the number of married people increased year by year, while the ratio of others (divorced, separated, widowed) decreased year by year. Patients from the Department of Internal Medicine (47.3%) and Department of Surgery (41.5%) were the main patients. Patients with (51.1%) and without (48.9%) undergoing surgery accounted for about half.

## Decreasing the pain prevalence rate after pain management

The hospital-wide pain management project was first implemented in 2018. As shown in Fig 2, the prevalence rates of pain in 2019 (63.3%) and 2020 (60.1%) were significantly lower than that of 2018 (69.5%) statistically. These prevalence rates between 2019 and 2020 did not reach a significant difference (OR: 0.872, p = 0.152). In terms of risk factors that cause pain (Table 2), male gender (OR: 0.821, p<0.05), elderly age (OR: 0.443, p<0.001), and existence of medical diseases (OR: 0.752, p<0.01) were correlated with lower prevalence of pain problem. The rate of pain in patients undergoing surgery was 2.136 times the rate of pain in patients without surgery (p<0.010). We also found that surgery was an important related factor for pain in each of the three consecutive years (OR:2.722, p<0.010) (OR:2.276, p<0.010) (OR:1.640, p<0.010).

## Pain management changed the patients' duration and severity of pain

Patients with pain during hospitalization (N = 719 in 2018, N = 655 in 2019 and N = 616 in 2020) reported the severity (Fig 3a) and duration (Fig 3b) of pain in the past 24 hours on the

**Table 2. The changes of risk factor among the patients with pain from 2018 to 2020.**

| Characteristics | ALL | 2018 | 2019 | 2020 |
|---|---|---|---|---|
| | N = 1990 (%) | N = 719 (%) | N = 655 (%) | N = 616 (%) |
| Gender | | | | |
| Female | 952 (47.8) | 353 (49.1) | 301 (46.0) | 298 (48.4) |
| Male | **1038 (52.2)**[a] | 366 (50.9) | 354 (54.0) | **318 (51.6)**[a] |
| Age (years) | | | | |
| ≤39 | 337 (16.9) | 123 (17.1) | 111 (16.9) | 103 (16.7) |
| 40~49 | **246 (12.4)**[a] | 88 (12.2) | **73 (11.1)**[a] | 85 (13.8) |
| 50~59 | 446 (22.4) | 174 (24.2) | 146 (22.3) | 126 (20.5) |
| 60~69 | **489 (24.6)**[a] | 175 (24.3) | **150 (22.9)**[a] | 164 (26.6) |
| 70~79 | **295 (14.8)**[a] | 95 (13.2) | **107 (16.3)**[a] | **93 (15.1)**[a] |
| ≥80 | **177 (8.9)**[a] | 64 (8.9) | **68 (10.4)**[a] | **45 (7.3)**[a] |
| Weight (kg) | | | | |
| ≤50 | 273 (13.7) | 102 (14.2) | 76 (11.6) | 95 (15.4) |
| 50~59 | 550 (27.6) | 194 (27) | 177 (27) | 179 (29.1) |
| 60~69 | 650 (32.7) | 223 (31) | **225 (34.4)**[a] | 202 (32.8) |
| 70~79 | 286 (14.4) | 120 (16.7) | 98 (15.0) | 68 (11.0) |
| ≥80 | 231 (11.6) | 80 (11.1) | 79 (12.1) | 72 (11.7) |
| Education | | | | |
| Illiterate | 131 (6.6) | 46 (6.4) | 48 (7.3) | 37 (6.0) |
| Elementary school | 391 (19.6) | 143 (19.9) | 145 (22.1) | 103 (16.7) |
| Junior high school | 296 (14.9) | 108 (15) | 99 (15.1) | 89 (14.4) |
| Senior high school | 545 (27.4) | 211 (29.3) | 164 (25.0) | 170 (27.6) |
| Bachelor's degree or higher | 627 (31.5) | 211 (29.3) | 199 (30.4) | 217 (35.2) |
| Marriage | | | | |
| Unmarried | 278 (14.0) | 100 (13.9) | 86 (13.1) | 92 (14.9) |
| Married | 1523 (76.5) | 543 (75.5) | 503 (76.8) | 477 (77.4) |
| Other | 189 (9.5) | 76 (10.6) | 66 (10.1) | 47 (7.6) |
| Disease categories | | | | |
| Surgery | 920 (46.2) | 320 (44.5) | 302 (46.1) | 298 (48.4) |
| Head, Neck, Ophthalmology | 120 (6) | 40 (5.6) | **44 (6.7)**[a] | 36 (5.8) |
| Internal Medicine | **803 (40.4)**[a] | **303 (42.1)**[a] | 258 (39.4) | 242 (39.3) |
| Gynecology | 147 (7.4) | 56 (7.8) | 51 (7.8) | 40 (6.5) |
| Operation | | | | |
| No operation | 805 (40.5) | 276 (38.4) | 257 (39.2) | 272 (44.2) |
| With operation | **1185 (59.5)**[a] | **443 (61.6)**[a] | **398 (60.8)**[a] | **344 (55.8)**[a] |

N (%): Data are expressed as the case number and its percentage among total cases.

[a]Odds ratio of multiple logistic regression for pain or severe pain significance (p< 0.05).

day of interview. We used a Numerical Rating Scale (NRS) to divide patients with pain into 4 groups: no pain (0), mild pain (1–3), moderate pain (4–6), and severe pain (7–10). The proportion of patients who reported no pain was the highest in 2019 (n = 216, 33%), compared to 2018 (n = 49, 6.8%) and 2020 (n = 85, 13.8%). The rates of moderate (n = 124, 18.9%) and severe (n = 87, 13.3%) pain were greatly reduced in 2019 in comparison with rates in 2018 (n = 196, 27.3% in moderate pain and n = 312, 43.4% in severe pain). Although the rates of moderate and severe pain slightly increased in 2020 (n = 188, 30.5% in moderate pain and n = 139, 22.6% in severe pain), it was still lower than that in 2018 (Fig 3a). The duration of

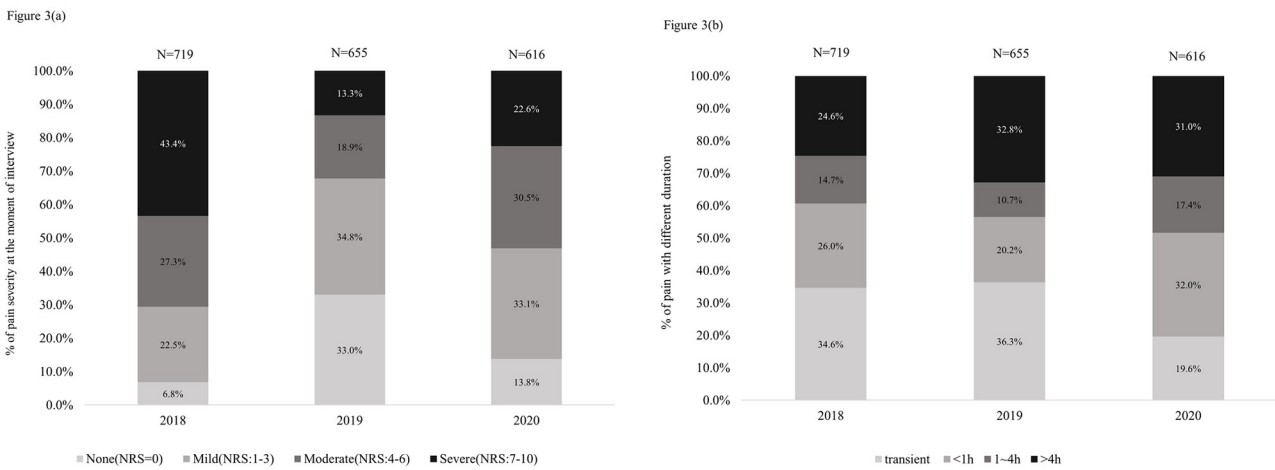

**Fig 3. The changes of pain severity (a) and pain duration (b) among the hospitalized patients with pain from 2018 to 2020.**

pain was divided into 4 groups: transient, less than 1 hour, 1–4 hours, and more than 4 hours. The results showed that the rate of those with longer pain duration increased in 2019 and 2020 in comparison with that in 2018 (Fig 3b).

## Care-related pain changed with pain management

For patients with moderate or severe pain, which lasted for more than 4 hours (N = 391), the number of cases and events are presented in Table 3. Injections (45%), wound dressing (31.7%), change of posture and chest percussion (21.0%) were the most common causes of pain. After the education and training of in-hospital medical staff, the rates of pain related to injections ($p<0.001$), change of posture and chest percussion ($p<0.001$), and rehabilitation ($p<0.05$) were lower in 2019 and 2020 compared with 2018.

## Discussion

The pain management project presented herein adopted an evidence-based approach with a data-driven information system to control patients' pain. We designed a novel indicator

**Table 3. The changes of care-related pain with severity $\geq 4$ and duration $\geq 4$ h.**

| Characteristics | ALL | 2018 | 2019 | 2020 | *p*-value |
|---|---|---|---|---|---|
|  | 391 (%) | 143 (%) | 119 (%) | 129 (%) |  |
| **Needle Pain** | **176 (45.0)** | **79 (55.2)** | **58 (48.7)** | **39 (30.2)** | **<0.001** |
| Wound dressing pain | 124 (31.7) | 43 (30.1) | 41 (34.5) | 40 (31.0) |  |
| **Change in posture/Chest percussion** | **82 (21.0)** | **46 (32.2)** | **14 (11.8)** | **22 (17.1)** | **<0.001** |
| Nasogastric tube | 17 (4.3) | 10 (7.0) | 4 (3.4) | 3 (2.3) |  |
| Foley Catheter | 22 (5.6) | 9 (6.3) | 7 (5.9) | 6 (4.7) |  |
| **Rehabilitation** | **22 (5.6)** | **10 (7.0)** | **10 (8.4)** | **2 (1.6)** | **<0.05** |
| Central venous catheter | 4 (1.0) | 2 (1.4) | 0 (0.0) | 2 (1.6) |  |
| Chest tube | 12 (3.1) | 3 (2.1) | 4 (3.4) | 5 (3.9) |  |
| Drainage tube | 1 (0.3) | 0 (0.0) | 1 (0.8) | 0 (0.0) |  |

We used the chi-square test to analyze changes in the proportions of the various causes of care-related pain over a three-year period.

termed "3324" on the dashboard, which was a real-time PDCA response cycle for the control of pain. After implementing the hospital-wide pain management project, we noted that the prevalence rate of pain decreased from 69.5% to 60.1% over the 3-year period of the study. Though the percentage of patients with long duration of pain was still high, the prevalence rates of pain and the rate of severe pain decreased. As in previous studies, our results indicate that pain remains a major problem in hospitalized patients [23–27]. The results indicate that this project was effective in moving "Towards a pain free-hospital".

How did this project improve hospital-wide pain management? A few factors contributed to the success of the project. First, we organized a multidisciplinary team and ran the regular activities of the team on wards. Leadership seemed to be pivotal to the success of the project. Second, we conducted a comprehensive education program in all wards and divisions. The educational content was evidence-based and updated every year, taking the shortcomings noted in the previous year into account. For example, we emphasized the importance of care-related pain (rehabilitation and change of posture). We found the rates of rehabilitation-related pain and change of posture-related pain decreased year by year. For pain specialists, our regular training activities are as follows. (1) Case discussion meetings were held regularly every month. (2) Pain physician instructed and demonstrated ultrasound-guided nerve block every week in the operation room. (3) Trainees, resident doctors and interns, participate in weekly pain medicine classes. The content of the training includes pharmacology, differential diagnosis of pain, techniques of regional analgesia, etc. For clinical staff, our training programs are as follows. We previously lacked standardized pain education and training. Since the Towards a Pain-free Hospital project, we began to develop painless education and training materials. In May 2018, the first edition of the pain teaching material was drawn up by Dr. Chih-Cheng Wu, the chief of the pain department. In June 2018, to train seed instructors, one from each department and two from each ward were assigned to attend the training. Afterward, these seed instructors returned to the department and ward for the first advocacy. In July 2018, pain physicians provided pain-free hospital guidance to all departments of the hospital, and the guidance was completed before the end of the year. Afterwards, our pain physicians hold regular lectures in various departments every year. Our guidance covers pharmacology, pain assessments, and dashboard pain management systems. Third, we applied the concept of continuous quality management using a real-time "3324" PDCA response cycle. Fourth, our pain management protocol was fully supported by the Data-driven Information Dashboard. The system closely integrated the roles of physicians, nurses, head nurses, and case managers.

The project did not change the risk factors of pain among the hospitalized patients, including gender, age, and surgical intervention. The prevalence of pain in females was higher than that in males in our study, which is consistent with the results of many previous studies [1, 24, 28–39]. Falk et al. explained that this may be explained, at least in part, by aspects of traditional culture and gender traits. Males are expected to be brave and may therefore tend to tolerate pain. In contrast, women may be deemed more likely to express emotion and be less able to endure pain compared with men [39].

In our study, the younger the age, the higher was the prevalence of pain. This finding is in line with previously reported results [1, 25, 28, 30–32, 40]. A study by Gianni et al. investigated elderly patients in eight Italian hospitals and also found that compared with young people, elderly patients did not report their pain problems because they believed that medical staff were busy with clinical work. If they reported pain, they feared that the medical staff would be unhappy with them [23]. The elderly have limited ability to express pain, so the proportion of actual pain remains high. Therefore, the pain problem of the elderly population cannot be underestimated and ignored.

The pain rate of patients in the department of surgery (n = 920, N = 1284, 71.7%) was significantly higher than that of patients in the department of internal medicine (n = 803, N = 1484, 54.8%), and patients who underwent surgery (n = 1185, N = 1580, 75.0%) had a higher rate of pain than patients who did not undergo surgery (n = 805, N = 1514, 53.2%) (Tables 1 and 2). This is consistent with the findings of Gregory et al. [3]. A study by Bourdillon et al. showed that in surgical patients, surgery was a key factor affecting pain [36]. In a survey conducted by Ambrogi et al. in a French university hospital, among 907 patients undergoing surgery, 330 (37%) experienced severe pain. The reason for this was presumed to be the decline of the anesthetic effect in surgical patients and the pain of wound treatment [26]. Other studies on the pain of inpatients showed similar results [24, 28, 29, 31, 36]. These results indicate that surgery is a powerful pain-causing factor. After three years of follow-up, we found that the pain rate of surgical patients decreased year by year. The changes in the pain rate of patients in the department of internal medicine indicate that our pain care measures had a further effect on perioperative pain.

The analysis of pain severity revealed that the proportion of moderate to severe pain decreased, and the proportion of mild pain increased relatively. The duration of pain findings indicate that the proportion of transient pain decreased, and the proportion of pain duration greater than 4 hours relatively increased. Our pain management system was particularly effective for perioperative pain. Postoperative wound pain was acute, and the degree was mostly moderate or severe. With immediate evaluation and treatment, pain can be relieved in a short time. However, the relative proportion of patients with chronic pain and internal medicine has increased. Most of these patients have pain before admission, and some are already receiving routine pain control. The pain of these patients is mostly mild (NRS: 1~3) and persistent. The 3324 system does not warn about this. This means that there is still room for improvement in the management of chronic pain in this system.

For persistent ($\geq$ 4 hours), moderate, and severe pain (NRS$\geq$4), we analyzed possible related factors in detail. Injection, changing the dressing of the wound, and change of position/chest percussion were the three most common care-related factors related to pain that were reported by patients. This finding is similar to the results of Coutaux et al. [41] and Ambrogi et al., [26] which showed injections and change in position/chest percussion were the most common causes of care-related pain in inpatients. We found that pain caused by injections, change in position/chest percussion, and rehabilitation had all been reduced. Among these factors, the rate of injection pain decreased significantly over the three consecutive years (49.3%, 42.1%, 27.9%) of this study. Taken together, these results indicate that the pain nursing education and training that we regularly organized in wards and departments might have improved the pain caused by the aforementioned factors.

There are some limitations within the study. First, our findings lack generalizability because all study populations were drawn from a single hospital. Second, we did not ask patients about pain prior to admission, nor did we investigate the detailed causes of pain. Third, we relied only on recalling the patient's past pain through questionnaires, and there is no objective recording tool in the process. Fourth, for patients who were illiterate or had difficulty understanding, the interviewer would personally ask and assist in answering. Although the research team conducted rigorous and standardized training of the interviewers before conducting the survey, inevitably there could have been discrepancies between the patients' actual feelings and their answers. Fifth, there was no difference in the demographic data during the three-year survey, except for age, weight, and marital status. Our sampling method tends to cause variability in the demographic data. We hope to reduce the bias, so we use logistic regression to analyze pain factors and pain prevalence trends. However, it may have influenced the results of the study. Sixth, we promoted Pain-Free Hospital in various units in our hospital every year.

Our guidance for clinicians includes pharmacology, pain assessments and dashboard pain management systems. In addition to promoting the above content, we provided new information on analgesics, additional tools for evaluating pain, and updates of the dashboard system. In order to promote the quality of Pain-Free Hospital, we updated the training content of medicine and informatics. Although standard process for inpatient pain management have not changed, it is possible that the updated information may have affected the results. Our study results represent the overall pain prevalence of inpatients. Despite the above limitations, this study not only provides a detailed description of the characteristics of pain prevalence in inpatients, but also sheds light on the effectiveness of the pain management system.

## Conclusions

Pain is a common complaint in all medical institutions. Our project effectively decreased the prevalence rate of pain among the hospitalized patients. The project included 4 core factors: (1) a multidisciplinary team, (2) an evidence-based education program, (3) a real-time "3324" PDCA response cycle to control pain, and (4) a data-driven Information Dashboard to integrate teamwork. Herein, we propose a model of pain management "toward a pain-free hospital". However, the results show the prevalence rate of pain up to 60% remains high. Hence, it is necessary to persevere in our efforts to improve the physical and mental condition of patients suffering from pain.

## Supporting information

**S1 Fig. The organizational structure of the pain-free hospital committee.**
(PDF)

**S2 Fig. 3324 Dashboard system.** We use the 3324 dashboard system to track patients' pain status and current prescriptions in real time, and pain experts could leave recommendations in the system after evaluation. This helped us improve the efficiency of pain management.
(PDF)

**S1 Table. Conference for review of pain management system.** Pain-free hospital group meetings are held regularly every two weeks to set up cross-team member participation, establish communication channels, and invite team members to discuss care implementation strategies. Monthly case discussions and ward team meetings were conducted for outpatient cases with special or refractory pain. The management committee discussed the business promotion, the work report of each group, the development of the annual policy, and the progress review on a quarterly basis, and reported the implementation progress to the hospital every six months.
(PDF)

**S1 Appendix. Structured questionnaire for patient perception of pain care survey.**
(PDF)

**S1 File. Questionnaire data from 2018 to 2020.**
(XLSX)

## Acknowledgments

We thank Hui-Hsun Chang, Ren-Jyun Yan, Ren-Qi Chen, Yi-Ting Liao, Ying-Jyun Shih and Yun-Tong Sun as interviewers. They helped us interview inpatients and collect questionnaires.

## Author Contributions

**Conceptualization:** Ming-Chuan Chen, Te-Feng Yeh, Chih-Cheng Wu, Chieh-Liang Wu, Ching-Hui Shen.

**Data curation:** Yan-Ru Wang, Ruei-ling Chen.

**Formal analysis:** Ruei-ling Chen, Ching-Hui Shen.

**Methodology:** Ming-Chuan Chen, Te-Feng Yeh, Yan-Ru Wang, Chieh-Liang Wu, Ching-Hui Shen.

**Project administration:** Ming-Chuan Chen.

**Supervision:** Ching-Hui Shen.

**Writing – original draft:** Ming-Chuan Chen, Chieh-Liang Wu.

**Writing – review & editing:** Ming-Chuan Chen, Te-Feng Yeh, Chih-Cheng Wu, Chieh-Liang Wu, Ching-Hui Shen.

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
