## [Decision Letter · Decision Letter 0]

6 Dec 2022

PONE-D-22-25681Three-year hospital-wide pain management system implementation at a tertiary medical center: pain prevalence analysisPLOS ONE

Dear Dr. Shen,

Thank you for submitting your manuscript to PLOS ONE. After careful consideration, we feel that it has merit but does not fully meet PLOS ONE’s publication criteria as it currently stands. Therefore, we invite you to submit a revised version of the manuscript that addresses the points raised during the review process. Please address each point raised by the reviewer and myself and provide your responses to each. Some feedback can be found in the annotated pdf file.

We look forward to receiving your revised manuscript.

Kind regards,

Christina Abdel Shaheed

Guest Editor

PLOS ONE

Journal Requirements:

Additional Editor Comments:

This is an interesting study however I have specific comments for the authors to consider. Please see annotations in the pdf file.

Reviewers' comments:

Reviewer's Responses to Questions

**Comments to the Author**

1. Is the manuscript technically sound, and do the data support the conclusions?

Reviewer #1: Yes

2. Has the statistical analysis been performed appropriately and rigorously? 

Reviewer #1: Yes

3. Have the authors made all data underlying the findings in their manuscript fully available?

Reviewer #1: Yes

4. Is the manuscript presented in an intelligible fashion and written in standard English?

Reviewer #1: Yes

5. Review Comments to the Author

Reviewer #1: Dear Team, thankyou for the opportunity to review the submission. The manuscript looked at evaluating the effect of a hospital wide pain management campaign and collected outcomes between 2018-2020. The details and information within the manuscript were presented in a sequence manner to describe the project design and the findings. There are a few minor comments to provide to see if these could be potential suggestions that could be reviewed:

Abstract:

Not to certain if this may be a typo, but a minor revision is required: The prevalence 28 rates of pain experienced by the patients in the 3-year period were 69.5% (2018) (reference)

Methods:

-Exclusion criteria: could you provide some examples of critical illness (e.g. cancer, heart/renal failure, patients receiving chemotherapy) to help better understand those that were not suitable for the study. Was pregnancy also excluded for the trial?

-Monitoring of those administering/reviewing the pain management program: This does not require an amendment to the manuscript and/or study design, but be keen to hear comments if there would be scope to undertake check-ups on process (e.g. 3 monthly) together with the 1 year that was initially undertaken for the study.

Results:

Demographics-

Disease categories. Were any of the participants enrolled into the study present with multiple disease categories? it may be worthwhile to see if this could impact on the response rates for the survey

Ethnicity- Not listed in the demographics but be good to see the population being assessed

Results:

Was the prevalence rates (pain) 2020 (60.1%) with inclusion of numbers during covid? Did the researchers feel the management was affected during the covid period as the rates of moderate severe pain increased in 2020? This would be understandable given the covid pandemic and the challenges the health practitioners may have experienced to manage an influx of patients during this period.

6. PLOS authors have the option to publish the peer review history of their article (what does this mean?). If published, this will include your full peer review and any attached files.

Reviewer #1: No

---

## [Author Response · Author response to Decision Letter 0]

18 Jan 2023

In email, Review#1 Comments to the Author

Abstract:

Not to certain if this may be a typo, but a minor revision is required: The prevalence 28 rates of pain experienced by the patients in the 3-year period were 69.5% (2018) (reference)

We changed it to “During the three-year period, the prevalence of pain was 69.5% (2018) (reference), 63.3% (2019) (OR:0.768, p<0.01), and 60.1% (2020) (OR:0.662, p <0.001).” Thank you for your advice.

Methods:

-Exclusion criteria: could you provide some examples of critical illness (e.g. cancer, heart/renal failure, patients receiving chemotherapy) to help better understand those that were not suitable for the study. Was pregnancy also excluded for the trial?

This is a misapplication of the wording and should be replaced by unstable medical condition, that refers to a state in which the condition is so unstable that pain cannot be clearly expressed, such as shock, coma, etc. Therefore, patients with pregnancy, cancer, and cardiovascular disease were not excluded unless they were unable to express themselves. Thanks for your helpful comments.

-Monitoring of those administering/reviewing the pain management program: This does not require an amendment to the manuscript and/or study design, but be keen to hear comments if there would be scope to undertake check-ups on process (e.g. 3 monthly) together with the 1 year that was initially undertaken for the study.

Some of our Pain-Free Hospital Management Committee comments on the program are as follows.

(1) Medical staff usually say to male patients that "pain is normal and must be tolerated". Medical staff need to minimize stereotypes. We should face up to everyone’s pain problem.

(2) Patients with advanced age and low education level are more difficult to use the numerical rating scale. It is recommended that medical staff choose appropriate assessment tools for different patients.

(3) Patients with severe pain and pain from injection and blood drawing are high-risk groups of side effects after taking analgesics. Medical staff should pay attention to the occurrence of side effects after taking analgesics.

(4) The interface of the information system should be tested and optimized as early as possible, so that the subsequent guidance and application will be smoother.

Results:

Demographics-

Disease categories. Were any of the participants enrolled into the study present with multiple disease categories? it may be worthwhile to see if this could impact on the response rates for the survey

Ethnicity- Not listed in the demographics but be good to see the population being assessed

This is a typo, and we changed it to “Medical department”, which refers to the department that provides primary care while hospitalized. This study did not investigate the patient's disease and race, and we hope to work on this part in the future.

Results:

Was the prevalence rates (pain) 2020 (60.1%) with inclusion of numbers during covid? Did the researchers feel the management was affected during the covid period as the rates of moderate severe pain increased in 2020? This would be understandable given the covid pandemic and the challenges the health practitioners may have experienced to manage an influx of patients during this period.

Because of COVID-19 outbreak, part of the ward was converted into a negative pressure isolation room. That part of the ward was excluded from sampling at the time, so 1,025 samples were collected in 2020 (N=1034 in 2018 and N=1035 in 2019). At that time, the workload of medical staff increased, so the work of pain management may be relatively neglected. We think this may be related to the elevated prevalence of moderate to severe pain in 2020.

Thank you for giving us the opportunity to strengthen our manuscript with your valuable comments and queries. We attach importance to your feedback and hope that these revisions persuade you to accept our submission.

Sincerely,

Dr. Ming-Chuan Chen

chen510156@gmail.com

---

## [Editor Report · Decision Letter 1]

24 Jan 2023

PONE-D-22-25681R1Three-year hospital-wide pain management system implementation at a tertiary medical center: pain prevalence analysisPLOS ONE

Dear Dr. Shen,

Thank you for submitting your manuscript to PLOS ONE. After careful consideration, we feel that it has merit but does not fully meet PLOS ONE’s publication criteria as it currently stands. Therefore, we invite you to submit a revised version of the manuscript that addresses the points raised during the review process.

It is important to describe the patient population more clearly. You refer to cancer guidelines, but were all the patients cancer patients? The management of cancer and non-cancer pain are quite different. Was the model you used applies to both groups of patients and if so, what is the justification for this?

We look forward to receiving your revised manuscript.

Kind regards,

Christina Abdel Shaheed

Guest Editor

PLOS ONE
---

## [Author Response · Author response to Decision Letter 1]

4 Mar 2023

Dear reviewer:

Thank you for inviting us to submit a revised draft of our manuscript entitled, “Three-year hospital-wide pain management system implementation at a tertiary medical center: pain prevalence analysis” to PLoS One. We also appreciate the time and effort you have dedicated to providing insightful feedback on ways to strengthen our paper. Thus, it is with great pleasure that we resubmit our article for further consideration. We have incorporated changes that reflect the detailed suggestions you have graciously provided. We also hope that our edits and the responses we provide below satisfactorily address all the issues and concerns you have noted.

To facilitate your review of our revisions, the following is a point-by-point response to the questions and comments delivered in your letter.

1. It is important to describe the patient population more clearly.

The detailed description of patient population in the article is as follows.

This study is a three-year longitudinal survey. All inpatients in general wards of Taichung Veterans General Hospital were the research subjects. The patients were enrolled if they (1) were at least twenty years old, (2) had a hospital stay of at least three days, (3) had clear consciousness, and (4) provided consent to be included in this study. Patients were excluded if they were not able to express themselves e.g., had dementia, brain injury, or unstable medical condition, such as shock or unconsciousness.

In total, 1034, 1035, and 1025 patients were recruited by quota sampling from wards in 2018, 2019, and 2020, respectively. A total of 3094 inpatients completed the survey in this study, including 1679 males (54.3%) and 1415 females (45.7%). They were divided into six groups according to age, among which 789 patients (25.5%) were aged 60-69. They were divided into five groups according to body weight, among which 966 patients (31.2%) accounted for 60-69 kg. 898 patients (29.0%) had a college education or above, followed by senior high school (27.4%), elementary school (21.8%), junior high school (14.9%), and illiterate (6.9%). In terms of marital status, most of the 2380 (76.9%) were married, and the number of married people increased year by year, while the ratio of others (divorced, separated, widowed) decreased year by year. Patients from the Department of Internal Medicine (47.3%) and Department of Surgery (41.5%) were the main patients. Patients with (51.1%) and without (48.9%) undergoing surgery accounted for about half.

2. You refer to cancer guidelines, but were all the patients cancer patients? The management of cancer and non-cancer pain are quite different. Was the model you used applies to both groups of patients and if so, what is the justification for this?

In response to the Towards a Pain-Free Hospital policy, considering the resources of the hospital and the feasibility of the policy, we hoped to develop a set of methods to cooperate with the dashboard system to monitor and identify patients with difficulty in pain management. Thus, we referred to the 3-3-3 principle and designed an approach termed “3324” for all pain monitoring in the hospital. This number means that the scale of VAS was > 3 and occurred ≥ 3 times within the last 24 hours during hospitalization. The data of patients who meet the criteria of 3324 will be displayed in the Dashboard Pain Management System, and the nurses and physicians can see it on the working panel of the electronic medical record system. The medical staff then take the necessary actions to control the patients’ pain following the process. If the patient still has refractory pain, the case manager of pain (a nurse) can visit the patient and consult with a pain specialist (anesthesiology) to adjust the analgesic prescription or consider regional analgesic therapies, such as peripheral nerve block. We hope to use this to identify inpatients who have difficulty in pain management immediately, and consult pain physicians to provide individualized treatment based on the patient’s condition, so as to improve the quality of pain management in the hospital.

Thank you for giving us the opportunity to strengthen our manuscript with your valuable comments and queries. We attach importance to your feedback and hope that these revisions persuade you to accept our submission.

Sincerely,

Dr. Ming-Chuan Chen

chen510156@gmail.com

---

## [Editor Report · Decision Letter 2]

12 Mar 2023

Three-year hospital-wide pain management system implementation at a tertiary medical center: pain prevalence analysis

PONE-D-22-25681R2

Dear Dr. Shen,

We’re pleased to inform you that your manuscript has been judged scientifically suitable for publication and will be formally accepted for publication once it meets all outstanding technical requirements.

Kind regards,

Christina Abdel Shaheed

Guest Editor

PLOS ONE

Additional Editor Comments (optional):

Thank you for your responses to my query. I think the responses are satisfactory.
---

## [Editor Report · Acceptance letter]

3 Apr 2023

PONE-D-22-25681R2 

Three-year hospital-wide pain management system implementation at a tertiary medical center: pain prevalence analysis 

Dear Dr. Shen:

I'm pleased to inform you that your manuscript has been deemed suitable for publication in PLOS ONE. Congratulations! Your manuscript is now with our production department. 

Kind regards, 

on behalf of

Dr Christina Abdel Shaheed 

Guest Editor

PLOS ONE